

# Separation of biospheric and fossil fuel fluxes of $CO_2$ by atmospheric inversion of $CO_2$ and $^{14}CO_2$ measurements: Observation System Simulations

Sourish Basu[1,2], John Bharat Miller[1,2], and Scott Lehman[3]

[1]Global Monitoring Division, NOAA Earth System Research Laboratory, Boulder CO, USA
[2]Cooperative Institute for Research in Environmental Science, University of Colorado, Boulder CO, USA
[3]Institute for Arctic and Alpine Research, University of Colorado Boulder, Boulder CO, USA

*Correspondence to:* Sourish Basu (sourish.basu@noaa.gov)

**Abstract.** National annual total $CO_2$ emissions from combustion of fossil fuels are likely known to within 5-10% for most developed countries. However, uncertainties are inevitably larger (by unknown amounts) for emission estimates at regional and monthly scales, or for developing countries. Given recent international efforts to establish emission reduction targets, independent determination and verification of regional and national scale fossil fuel $CO_2$ emissions are likely to become increasingly important. Here, we take advantage of the fact that precise measurements of $^{14}C$ in $CO_2$ provide a largely unbiased tracer for recently added fossil fuel derived $CO_2$ in the atmosphere and present an atmospheric inversion technique to jointly assimilate observations of $CO_2$ and $^{14}CO_2$ in order to simultaneously estimate fossil fuel emissions and biospheric exchange fluxes of $CO_2$. Using this method in a set of Observation System Simulation Experiments (OSSEs), we show that given the coverage of $^{14}CO_2$ measurements available in 2010 (969 over North America, 1063 globally), we can recover the US national total fossil fuel emission to better than 1% for the year and to within 5% for most months. Increasing the number of $^{14}CO_2$ observations to ~5,000 per year over North America, as recently recommended by the National Academy of Science (NAS) (Pacala et al., 2010), we recover monthly emissions to within 5% for all months for the US as a whole and also for smaller, highly emissive regions over which the specified data coverage is relatively dense, such as for the New England states or the NY-NJ-PA tri-state area. This result suggests that, given continued improvement in state-of-the art transport models, a measurement program similar in scale to that recommended by the NAS can provide for independent verification of bottom-up inventories of fossil fuel $CO_2$ at the regional and national scale. In addition, we show that the dual tracer inversion framework can detect and minimize biases in estimates of the biospheric flux that would otherwise arise in a traditional $CO_2$-only inversion when prescribing fixed but inaccurate fossil fuel fluxes.

## 1 Introduction

The terrestrial biosphere and the oceans have taken up roughly half the anthropogenic emissions of $CO_2$, with the remainder contributing to the observed increase in atmospheric $CO_2$ concentration from ~280 ppm in the early 1800s to ~395 ppm in 2013 (Ballantyne et al., 2012). But while $CO_2$ observations from sampling networks over large, industrialized land areas will





be influenced by emissions from combustion of fossil fuels, they are often dominated by seasonally and diurnally varying fluxes of the terrestrial biosphere. Thus, it is nearly impossible to make use of the atmospheric $CO_2$ observations alone as an independent constraint on the space-time patterns of fossil fuel $CO_2$ emissions (Shiga et al., 2014). In addition, conventional inversion schemes (Rödenbeck et al., 2003; Peters et al., 2007; Gurney et al., 2008; Mueller et al., 2008; Chevallier et al.,

2010a; Basu et al., 2013; Takagi et al., 2014) typically prescribe fossil fuel $CO_2$ fluxes from inventories based on economic statistics on fossil fuel consumption and assumed combustion efficiencies (Andres et al., 2012) with an assigned uncertainty of zero. Under these conditions, any deviation of the prescribed fossil fuel $CO_2$ fluxes from their true values can be expected to result in errors in the retrieved estimates of the terrestrial biospheric exchange fluxes. In areas over which the total carbon budget is well constrained by large number of observations, such as the conterminous US, these "carry-over biases" may be

comparable in magnitude to the errors in the specified fossil fuel $CO_2$ fluxes themselves.

The assumption of perfectly well-known fossil fuel fluxes has been a reasonable starting point since annual total fossil fuel $CO_2$ emissions from most developed (i.e., UNFCCC "Annex I" and "Annex II") countries are likely known to within 5% (Andres et al., 2012), a level of certainty that greatly surpasses our knowledge of the annual net terrestrial biosphere $CO_2$ flux over those areas. However, for developing (non-Annex) countries, fossil fuel uncertainties are likely to be much larger. For

example, estimates of Chinese emissions from fossil fuel combustion and cement production have been revised by +17% (Guan et al., 2012) and -14% (Liu et al., 2015) over the past five years alone. Moreover, uncertainties in fossil fuel $CO_2$ emissions are likely to be larger (by unknown amounts) at sub-national and sub-annual scales, even in developed countries. To illustrate this, we show maps of the difference between two widely used inventories of the annual fossil fuel $CO_2$ flux over North America along with an estimate of annual average net ecosystem exchange (NEE) in Figure 1. The inventory differences are in some

cases similar in magnitude to estimates of NEE for individual grid cells (at a resolution of $1° \times 1°$ in this example). Making matters worse, it is frequently necessary to extrapolate emissions inventories forward in time to correspond with the times of atmospheric observations. Such extrapolations might reasonably account for changes in population but will not capture changes in fossil fuel use associated with, for example, protracted regional heat and cold waves. At the time of this writing, both the Vulcan (http://vulcan.project.asu.edu/research.php) and EDGAR (http://edgar.jrc.ec.europa.eu/overview.php?v=42)

inventories provide emissions estimates only up to 2008, and even the "fast track" version of EDGAR (EDGAR v4.2 FT2010) has yet to be updated beyond 2010.

Here we take a first step at determining fossil fuel emissions using an atmospheric "top-down" method and evaluating our ability to reduce carry over bias by making use of the existing and anticipated array of precise measurements of atmospheric $^{14}CO_2$, which provide for direct, precise (~1 ppm) and largely unbiased observational constraints on fossil fuel derived $CO_2$

in the same samples that provide the primary $CO_2$ observations (Turnbull et al., 2009; Levin et al., 2011; Miller et al., 2012; Lehman et al., 2013). Below we first describe a new inversion framework that assimilates both $CO_2$ and $^{14}CO_2$ in a system that simultaneously optimizes both fossil fuel and biospheric exchange fluxes of $CO_2$. We then outline a set of Observation System Simulations Experiments (OSSEs) designed to evaluate the ability of the dual-tracer inversion framework to separately estimate these fluxes over the conterminous US using synthetic observations corresponding in space and time to (a) actual

observations in the NOAA ESRL Greenhouse Gas Reference Network in 2010 (1063 $^{14}CO_2$ measurements globally, of which





969 were in North America) and (b) an enhanced observational network with 6448 $^{14}CO_2$ measurements globally in 2010 (5304 in North America), similar to the annual $^{14}CO_2$ coverage recently recommended by the US NAS (Pacala et al., 2010). We use the observational network of (b) in three additional experiments. First, we perform an ensemble of inversions with and without $^{14}CO_2$ data in order to evaluate the degree to which the inclusion of $^{14}CO_2$ observations allows us to distinguish

between biospheric and fossil fuel $CO_2$ fluxes. We also repeat (b) without $^{14}CO_2$ data in order to quantify (by contrast to the dual tracer results) the degree to which the dual tracer system is able to detect and minimize potential carry over bias in NEE that might otherwise arise from a biased fossil fuel prior. Finally we repeat (b) but with different models of atmospheric transport to generate and assimilate the synthetic observations, in order to evaluate the potential impact of transport model error on our emissions estimates.

## 2 The inversion framework

Our inversion framework builds on the existing TM5 4DVAR system (Meirink et al., 2008), which has been used for estimating sources and sinks of $CH_4$ (Bergamaschi et al., 2013; Houweling et al., 2014), CO (Hooghiemstra et al., 2011), $CO_2$ (Basu et al., 2013) and $N_2O$ (Corazza et al., 2011). Here we describe modifications to the TM5 4DVAR system that permit us to jointly assimilate the measurements of two tracers, $CO_2$ and $^{14}CO_2$.

The atmospheric mass balances of $CO_2$ and $^{14}CO_2$ have been presented previously by Miller et al. (2012). Following those equations, we rewrite the isotopic mass balance (equations 1b and 1c of Miller et al. (2012)) in terms of the transported and conserved quantity $C\Delta_{atm}$, while the carbon balance (equation 1a) remains the same, such that:

$$\frac{d}{dt}C = F_{bio} + F_{oce} + F_{fos} \tag{1a}$$

$$\frac{d}{dt}(C\Delta_{atm}) = \frac{N}{r_{std}}(F_{nuc} + F_{cosmo}) + \Delta_{fos}F_{fos} + \Delta_{atm}(F_{oce} + F_{bio})$$

$$+ (\Delta_{oce} - \Delta_{atm})F_{oceatm} + (\Delta_{bio} - \Delta_{atm})F_{bioatm} \tag{1b}$$

$$= \frac{N}{r_{std}}(F_{nuc} + F_{cosmo}) + \Delta_{fos}F_{fos} + \Delta_{atm}(F_{oce} + F_{bio})$$

$$+ F_{ocedis} + F_{biodis} \tag{1c}$$

$C$ is the atmospheric burden of $CO_2$, while $\Delta_{atm}$ is the isotope signature of $^{14}CO_2$ in the atmosphere expressed in $\Delta$ notation,

which includes corrections for mass dependent isotopic fractionation between reservoirs and radioactive decay between the times of sample collection and measurement, such that the quantity $\Delta^{14}CO_2$ is conserved in time (Stuiver and Polach (1977), where $\Delta^{14}C$ is equivalent to $\Delta^{14}CO_2$ here). $F_{bio}$, $F_{oce}$ and $F_{fos}$ are net $CO_2$ surface fluxes to the atmosphere from the terrestrial biosphere, oceans and fossil fuel burning respectively, and we set $\Delta_{fos}$ to -1000 ‰, corresponding to a fossil fuel source devoid of $^{14}C$ as a result of radioactive decay. $F_{nuc}$ is the $^{14}CO_2$ flux from nuclear power and reprocessing plants, and $F_{cosmo}$ is the

cosmogenic production of $^{14}CO_2$, corresponding to the terms $isoF_{nuc}$ and $isoF_{cosmo}$ respectively of Miller et al. (2012). To





convert these pure $^{14}CO_2$ fluxes into units of $CO_2$ flux $\times \Delta$ (e.g., PgC/yr/‰), as in the other terms on the right hand side of equation (1c), we divide by the $^{14}$C:C standard ratio, $r_{std} = 1.176 \times 10^{-12}$, and account for mass dependent fractionation by multiplying by $N = (975/(\delta^{13}C + 1000))^2$ (Stuiver and Polach, 1977), where $\delta^{13}C$ has an assumed atmospheric value of $-8‰$. $\Delta_{oce}$ and $\Delta_{atm}$ are the isotope signatures of the ocean and the atmosphere respectively. In equation (1c) we assume that in converting from $^{14}$C:$^{12}$C to $\Delta^{14}$C all isotopic fractionation between reservoirs "drop out" of the equations, such that we can equate the isotopic signature $\Delta_{atm \to x}$ to $\Delta_{atm}$, and $\Delta_{x \to atm}$ to $\Delta_x$. $F_{oceatm}$ and $F_{bioatm}$ are the one-way gross ocean to atmosphere and biosphere to atmosphere $CO_2$ fluxes. The terms $F_{ocedis} = (\Delta_{oce} - \Delta_{atm})F_{oceatm}$ and $F_{biodis} = (\Delta_{bio} - \Delta_{atm})F_{bioatm}$ are so-called disequilibrium fluxes (where $\Delta_{bio} - \Delta_{atm} = \Delta_{biodis}$ in Miller et al. (2012)). Note, finally, that an extra term involving the net ocean and terrestrial fluxes ($F_{oce}$ and $F_{bio}$) appears in equation (1c), compared to (1b) of Miller et al. (2012), due to the slightly different left hand sides ($d(C\Delta_{atm})/dt$ vs $Cd\Delta_{atm}/dt$) of the two equations. Their magnitudes are only $\sim$ $100\,\text{PgC/yr/‰}$ which is relatively small compared to, for example, the fossil fuel flux of $\sim$10,000 PgC/yr/‰.

To solve equations (1) in an inversion, we further separate terms in equation (1a) into the sum of oceanic and terrestrial biospheric (i.e., "natural") components and fossil fuel components, where $CO_2^{ff}$ denotes the $CO_2$ in the atmosphere accumulated due to fossil fuel burning since the beginning of the simulation period ($t_0$).

$$\frac{d}{dt}CO_2^{nat} = F_{oce} + F_{bio} \tag{2a}$$

$$\frac{d}{dt}CO_2^{ff} = F_{fos} \tag{2b}$$

$$CO_2^{ff}(t = t_0) = 0 \tag{2c}$$

Our system is primarily designed to estimate fossil fuel $CO_2$ fluxes and NEE. However, we also solve for $F_{ocedis}$ and $F_{biodis}$ at a coarser temporal resolution, as explained in § 2.1.2. Note that equation (1c) contain $\Delta_{atm}$ on both sides. However, we do not solve for a $\Delta_{atm}$ field self-consistently within the inversion framework. On the left hand side, we treat $C\Delta_{atm}$ as a single tracer. Accordingly, we convert all measured $^{14}CO_2$ values to "measurements" of $C\Delta_{atm}$ for the flux estimation. On the right hand side, for the term $\Delta_{atm}(F_{oce} + F_{bio})$, we specify a $\Delta_{atm}$ that is spatially uniform and has a smooth temporal variation based on observations from the well mixed free troposphere at Niwot Ridge, Colorado (NWR: 40.0531°N, 105.5864°W, http://www.esrl.noaa.gov/gmd/dv/iadv/graph.php?code=NWR&program=ccgg&type=ts), filtered to remove possible local urban influences from the Denver-Boulder area to the east (Lehman et al., 2013). The error made in the inversion by using this smoothed approximation of $\Delta_{atm}$ on the right hand side of equation (1c) is small, since it will in practice be very close to $\Delta_{atm}$ in $C\Delta_{atm}$ and, as noted above, the term $\Delta_{atm}(F_{oce} + F_{bio})$ is small compared to others in the overall budget. For the disequilibrium fluxes on the right hand side, we solve for $F_{ocedis}$ and $F_{biodis}$ but do not attempt to separate those into the one-way gross $CO_2$ fluxes and their respective isotopic disequilibria.



### 2.1 Modeling framework

#### 2.1.1 Atmospheric transport

We use the TM5 atmospheric tracer transport model (Krol et al., 2005) to simulate atmospheric tracer concentrations from surface fluxes. TM5 can be run with convective entrainment and detrainment fluxes determined directly from the ERA-interim

reanalysis from the European Center for Medium range Weather Forecast (henceforth called "TM5 EIC") or with those fluxes computed within TM5 according to the convective scheme of Tiedtke (1989) (henceforth, "TM5 EI"), which was the standard scheme prior to 2014. The largest difference between "TM5 EI" and "TM5 EIC" is in the vertical transport into the free troposphere over temperate latitudes. For tracers with surface sources and sinks and negligible atmospheric chemical production and loss – such as $CO_2$ and $SF_6$ – this difference creates markedly different north-south (N-S) gradients at the surface, even

though the advective winds are the same. As an illustration, we show in Figure 2 the average simulated N-S gradient of $SF_6$ within the marine boundary layer for both TM5 EIC and TM5 EI, compared to average observations from 2002-2011.

The 0.3 ppt N-S gradient in $SF_6$ of TM5 EIC is very close to the observed gradient of 0.295 ppt, whereas the 0.38 ppt N-S gradient of TM5 EI is the farthest outlier among sixteen global transport models considered by Patra et al. (2011). Moreover, in the analysis of Patra et al. (2011), most modeled N-S gradients were between 0.27 ppt and 0.32 ppt. Thus, the difference of

15 0.08 ppt in the N-S gradients simulated by TM5 EI and TM5 EIC is larger than typical inter-model differences, indicating that these two schemes represent very different realizations of transport, at least at the hemispheric and global scales. Since TM5 EIC delivers markedly better agreement with the observed $SF_6$ N-S gradient, we use TM5 EIC for both forward simulation and inversion in all experiments, except when evaluating the impact of transport error on estimated fluxes (for which we use TM5 EI to assimilate synthetic observations produced by TM5 EIC, as outlined in § 3.4).

To better resolve atmospheric transport over the domain of interest, we run the atmospheric transport model at $1° \times 1°$ resolution over North America ($20°$ N to $64°$ N, $132°$ W to $60°$ W), and at $3° \times 2°$ resolution elsewhere. This is the same nested zoom configuration employed in NOAA's CarbonTracker North America.

#### 2.1.2 TM5 4DVAR

The TM5 4DVAR inversion system estimates fluxes $\mathbf{x}$ given observations $\mathbf{y}$ by minimizing the so-called cost function $J$

(Meirink et al., 2008),

$$J = \frac{1}{2}\left(H\mathbf{x} - \mathbf{y}\right)^T R^{-1}\left(H\mathbf{x} - \mathbf{y}\right) + \frac{1}{2}\left(\mathbf{x} - \mathbf{x}_0\right)^T B^{-1}\left(\mathbf{x} - \mathbf{x}_0\right) \tag{3}$$

where $H$ is an atmospheric transport operator, $\mathbf{x}_0$ is the prior flux before doing a data assimilation, and $R$ and $B$ are the respective error covariance matrices of the model data mismatch and the prior flux. The TM5 variational framework for atmospheric inversion of a single species has been described in detail previously (Meirink et al., 2008; Hooghiemstra et al., 2011; Basu

et al., 2013). In this work, $\mathbf{x}$ contains the surface fluxes of the three species $CO_2^{ff}$ ($F_{fos}$), $CO_2^{nat}$ ($F_{oce}$ and $F_{bio}$) and $C\Delta_{atm}$ ($F_{ocedis}$ and $F_{biodis}$). We solve for $F_{bio}$, $F_{oce}$, and $F_{fos}$ weekly, and for $F_{biodis}$ and $F_{ocedis}$ monthly. The prior flux error covariance



matrix is assumed to be separable in time and space, as in

$$B(\mathbf{r}_1,t_1;\mathbf{r}_2,t_2) = \mathrm{cov}(x_{\mathbf{r}_1,t_1}, x_{\mathbf{r}_2,t_2}) = \sigma_{\mathbf{r}_1,t_1}\sigma_{\mathbf{r}_2,t_2}C_{\mathbf{r}}(\mathbf{r}_1,\mathbf{r}_2)C_t(t_1,t_2) \qquad (4)$$

where $\mathbf{r}$ and $t$ are space and time coordinates respectively, $\sigma_{\mathbf{r},t}$ is the uncertainty of the prior flux at location $\mathbf{r}$ at time $t$, and $C_{\mathbf{r}}(\mathbf{r}_1,\mathbf{r}_2)$ (or $C_t(t_1,t_2)$) is the correlation between flux errors at locations $\mathbf{r}_1$ and $\mathbf{r}_2$ (or times $t_1$ and $t_2$). No prior correlation

is assumed between the five flux categories being optimized. For each category, the temporal error correlation $C_t$ is assumed to be exponential, $C_t(t_1,t_2) = e^{-|t_1-t_2|/T}$, with $T$ being three months for $F_{\mathrm{oce}}$, $F_{\mathrm{fos}}$ and $F_{\mathrm{ocedis}}$, and one month for $F_{\mathrm{bio}}$ and $F_{\mathrm{biodis}}$. The spatial error correlation is either

(a)  exponential, $C_{\mathbf{r}}(\mathbf{r}_1,\mathbf{r}_2) = e^{-|\mathbf{r}_1-\mathbf{r}_2|/L}$, or

(b)  "regional", where the globe is subdivided into regions, and grid cells within one region are perfectly correlated, whereas
grid cells from different regions are completely uncorrelated, or

(c)  a "hybrid" of the first two, where the grid cell to grid cell correlation decays exponentially within each defined region, but is zero between regions.

We denote these three types of spatial correlations **e**, **r** and **h** respectively. The parameters of spatial correlation for the five categories, as well as the per-grid cell prior errors (i.e., $\sigma_{\mathbf{r},t}$ of equation (4)) are listed in Table 2.

Surface fluxes are solved for at the same lateral resolution as the transport ($3° \times 2°$ globally, $1° \times 1°$ over N America), to provide the inversion with flexibility to change surface fluxes where there are more observations, and to reduce aggregation error (Kaminski et al., 2001). This transport/flux configuration is similar to NOAA's CarbonTracker North America (carbontracker.noaa.gov), except that we solve for additive corrections to per-grid cell surface fluxes instead of multiplicative corrections to regional surface fluxes. We focus on the year 2010, and our inversions run from July 4, 2009 to April 1, 2011,
to allow for sufficient spin up time at the beginning and sufficient time for the fluxes at the end of 2010 to be captured by subsequent observations.

## 2.2  $^{14}CO_2$ flux terms

Equations (1a) and (1c) contain seven different flux terms on the right hand side. In the OSSE we create synthetic observations of $C\Delta_{\mathrm{atm}}$ by specifying emissions fields and transporting "true" emissions for all seven terms. For the inversions, we specify
prior fluxes associated with fossil fuel $CO_2$ emissions ($F_{\mathrm{fos}}$) and net oceanic and biospheric fluxes ($F_{\mathrm{oce}}$ and $F_{\mathrm{bio}}$) that differ from those used to produce the simulated observations, and evaluate our ability to recover true fluxes using the synthetic observations. The two different sets ("true" vs. "a priori") of fossil fuel $CO_2$ and net $CO_2$ flux terms are described in § 3.3. The construction of the isofluxes for the remaining terms is described below and is consistent with the recent tropospheric $\Delta^{14}CO_2$ budget and distribution based on observations.
Gridded estimates of the $^{14}C$ production flux from nuclear reactors and fuel reprocessing plants, $F_{\mathrm{nuc}}$, were taken from Graven and Gruber (2011) and did not vary with time. Only the portion of this flux estimated to be directly emitted as $^{14}CO_2$



was included. The production of $^{14}C$ in the atmosphere, $F_{cosmo}$, and the sensitivity of this production to geomagnetic latitude depend on the solar modulation parameter $\Phi$, a scalar which varies with time. Annual values of $\Phi$ were calculated through 2012 based on a global array of neutron monitor data obtained from http://nmdb.eu/ (all amplitude normalized to count rates at Deep River, Canada, http://neutronm.bartol.udel.edu/~pyle/bri_table.html) and the slope of a linear regression between annual average Deep River Neutron Monitor count rate and estimates of $\Phi$ between 1955 and 1995 from Masarik and Beer (1999). Then, for each year of our simulation period, we calculated the $^{14}C$ production as a function of geomagnetic latitude given the annual average $\Phi$ of that year (Masarik and Beer, 2009). This resulted in annually varying production fields dependent on geomagnetic latitude. These production fields were then distributed vertically over the TM5 model layers corresponding to the stratosphere (between 150 hPa and 3 hPa), with the mass of $^{14}CO_2$ in each layer proportional to the total mass of air in that layer. To better match the observed $^{14}CO_2$ trend at Niwot Ridge, Colorado (NWR), the global total cosmogenic production was scaled by 0.9 in all years.

To calculate the terrestrial disequilibrium flux term $(\Delta_{bio} - \Delta_{atm})F_{bioatm}$, we first constructed the historical time series of atmospheric $^{14}CO_2$ by compositing overlapping time series from tree ring measurements (Stuiver, 1982), atmospheric records from Vermunt (Levin et al., 1994), Schauinsland $^{14}C$ (Levin and Kromer, 1997), Jungfraujoch $^{14}C$ (Levin et al., 2013), and more recently Niwot Ridge (Lehman et al., 2013). This historical time series was convolved with the age distribution of respired carbon derived from pulse-response functions from the Carnegie Ames Stanford Approach (CASA) biosphere model (Thompson and Randerson, 1999), to obtain a monthly $\Delta^{14}CO_2$ of respired carbon, "$\Delta_{bio}$", for each $1° \times 1°$ CASA grid cell. "$\Delta_{atm}$" was derived from filtered, monthly average observations at Niwot Ridge, CO (NWR) to obtain $(\Delta_{bio} - \Delta_{atm})$, and $F_{bioatm}$ was determined from the monthly total heterotrophic respiration flux for each CASA grid cell. Monthly $F_{bioatm}$ did not vary from year to year, while $\Delta_{bio}$ and $\Delta_{atm}$ were updated monthly and from year to year based on observed changes in atmospheric $\Delta^{14}CO_2$.

The oceanic isotopic disequilibrium $(\Delta_{oce} - \Delta_{atm})$ was estimated from observations of the $\Delta^{14}C$ of surface ocean dissolved inorganic carbon field available from World Ocean Circulation Experiment (WOCE) for the 1980's – 1990's and updated yearly through 2012 using rates of change for different ocean regions based on subsequent observations from the Climate and Ocean Variability (CLIVAR) measurement program (http://cdiac.ornl.gov/oceans/datmet.html). The gridded annual estimates of $\Delta_{oce}$ were differenced from a zonally uniform surface layer $\Delta_{atm}$ field based on filtered and seasonally smoothed observations from Niwot Ridge, CO (NWR) but with a specified increase of $+10\,‰$ between 20°N and 20°S. The disequilibrium flux was then calculated by scaling the isotopic disequilibrium by the one-way ocean to atmosphere $CO_2$ flux for each grid cell, which was derived from a climatology of surface ocean $pCO_2$ from Takahashi (2009) and a quadratic windspeed-dependent piston velocity (Wanninkhof, 1992) scaled to a more recent analysis of the oceanic $^{14}C$ inventory (Sweeney et al., 2007).

## 2.3 Initial atmospheric $CO_2$ and $^{14}CO_2$ fields

Initial concentration fields of $CO_2$ and $^{14}CO_2$ for the inversions were obtained by specifying realistic troposphere-stratosphere and latitude gradients of $\Delta^{14}CO_2$ and $CO_2$ and then propagating time-varying flux terms in equations (1) through the atmosphere using TM5, starting on Jan 1, 2000. The three dimensional atmospheric mole fractions of $CO_2 \times \Delta^{14}CO_2$ and $CO_2$ on



July 4, 2009 were used as initial fields for the inversions. The relatively long forward run was implemented to ensure that the simulated large-scale atmospheric gradients were consistent with the prior fluxes.

## 3 Experimental design

Our OSSEs (Table 1) are designed to evaluate the ability of a network of $^{14}CO_2$ observations – in conjunction with more widely available $CO_2$ observations – to constrain regional fossil fuel $CO_2$ and net biosphere exchange fluxes within our inversion framework. To do this, we first create synthetic atmospheric $^{14}CO_2$ and $CO_2$ concentrations at real and projected measurement locations based on transport of a set of "true" fluxes in TM5 EIC (this step is sometimes referred to as the "nature run" for an OSSE). By "true" we do not suggest that these fluxes are accurate but that they are consistent with the synthetic observations for the purpose of conducting the OSSE. We then assimilate the synthetic measurements in an atmospheric inversion using prior flux estimates which differ substantially from the "true" fluxes. The extent to which fluxes estimated by the inversion match the "true" fluxes is a measure of the performance of our inversion framework and the network of (synthetic) observations. An additional metric of performance is the degree to which $^{14}CO_2$ data can distinguish between NEE and fossil fuel $CO_2$ fluxes, measured by the posterior correlation between the two. This metric is further discussed in § 3.5 and § 4.1.

### 3.1 "True" fluxes

"True" fluxes used to simulate the observations are those for $^{14}CO_2$ described in § 2.2 along with those for fossil fuel $CO_2$ and net ocean and biosphere exchange. For fossil fuel $CO_2$, we use fossil fuel fluxes from CarbonTracker 2013, redistributed within the continental US according to the Vulcan spatiotemporal pattern. In addition, we impose scaling factors of Nassar et al. (2013) in order to represent the diurnal variability. "True" ocean fluxes were taken from posterior fluxes of CarbonTracker 2013b, specifically the variant which used the ocean interior inversion of Jacobson et al. (2007b) to construct prior ocean fluxes. "True" terrestrial fluxes were based on CASA GFED 3 (van der Werf et al., 2003). CASA GFED 3 provided only monthly NEE fluxes; in order to represent variability at higher frequencies we imposed daily and three hourly variations from SiB CASA GFED4 (van der Velde et al., 2014) on the monthly fluxes.

### 3.2 Synthetic observations

We simulated two sets of observations, with distributions as shown in Figure 4 and Table 3. The first set, which we refer to as "2010 coverage", placed a $^{14}CO_2$ (or $CO_2$) observation at each spatiotemporal point where there was an actual $^{14}CO_2$ (or $CO_2$) measurement between July 4, 2009 and April 1, 2011. This resulted in a total of 1,639 $^{14}CO_2$ and 45,330 $CO_2$ observations over the 21 month period (1,475 and 18,008 over North America, respectively). The accuracy of the estimated surface fluxes with respect to the "true" fluxes provided a measure of the performance of the real observational network in 2010.

For the second set, which we refer to as "NRC 5000", we simulated ~5,000 $^{14}CO_2$ measurements per year over North America. In constructing the expanded, hypothetical observational network (Figure 4) we first sought to increase measurements at existing NOAA and NOAA-partner monitoring locations, including tall towers and airborne and surface flask sampling



locations, adding six new tall tower sites to fill gaps in the sampling network. For $CO_2$ we also added shipboard samples from two monthly cruises in the Pacific Ocean. Table 3 lists the sampling frequencies for $CO_2$ and $^{14}CO_2$ at the different sites.

The design of the NRC 5000 network conformed as closely as possible to the actual sampling protocols and periodicities at tower, flask, aircraft and cruise locations maintained by NOAA and its partner networks. At tower sites, we sampled the "true"

$CO_2$ field at the highest intake height, at 00:30 and 03:30 local solar time (LST) for mountaintop sites and at 12:30 and 15:30 LST otherwise. The "true" $^{14}CO_2$ field was sampled on Mondays and Thursdays following the same protocol for intake height and LST. Flask sites were sampled on Wednesdays at 13:30 LST (01:30 LST for mountaintop sites) for both tracers. Some NOAA flask sites – such as Ascension Island, Cold Bay (Alaska) and Guam – collect $CO_2$ samples less frequently. At those sites, our sampling followed the protocol for $CO_2$ at the other flask sites, but with sampling only every other week. At aircraft

sites, we sampled simulated $CO_2$ at 13:30 LST, at altitudes where actual $CO_2$ samples are obtained (typically every 1000 to 2000 feet, to a site-dependent maximum altitude). This resulted in between nine and twelve samples per profile, depending on the site. For $^{14}CO_2$, three samples were taken per aircraft profile, distributed between the boundary layer and the free troposphere, reflecting the actual ongoing aircraft sampling strategy for $^{14}CO_2$ (c.f. Miller et al. (2012)). Shipboard samples for $CO_2$ were simulated as samples along a transect once every 5° latitude, successive samples being separated by one day, along

NOAA Pacific Ocean and Western Pacific cruises, which go back and forth once a month.

### 3.3 Prior fluxes for OSSE inversions

For the inversion of synthetic observations, we specified a set of prior fossil fuel $CO_2$ and net biospheric and oceanic fluxes that differed from those used to create the data. Prior fossil fuel $CO_2$ fluxes were taken from the EDGAR 4.2 FT2010 global inventory (http://edgar.jrc.ec.europa.eu/overview.php?v=42FT2010). EDGAR fluxes were available at $1° \times 1°$ resolution, but

had no sub-annual variability and were available only through 2010. For 2011, country totals for 2010 were scaled up according to the BP growth rate between 2010 and 2011 for each country (http://www.bp.com/en/global/corporate/about-bp/energy-economics/statistical-review-of-world-energy/statistical-review-downloads.html). The fossil fuel flux was optimized over weekly time steps. We imposed – but did not optimize – an hour-of-day variability on the fossil fuel fluxes using the diurnal (but not day of week) scaling factors of Nassar et al. (2013). Prior terrestrial fluxes were from SiBCASA/GFED4, which

included NEE, fires and biomass burning (van der Velde et al., 2014). The fluxes were specified globally on a $1° \times 1°$ grid at three hour time steps. The inversion optimized weekly terrestrial fluxes at the lateral resolution of the TM5 transport model. Within one week, the prior three-hourly variations were imposed as additive temporal patterns, but not optimized; i.e., only the mean NEE over a week was adjusted. Prior oceanic $CO_2$ fluxes, also at $1° \times 1°$ and three hourly resolution, were taken from the ocean prior of CarbonTracker 2013b (the variant based on Jacobson et al. (2007a)), and optimized weekly. The prior errors

assumed for the different fluxes are listed in Table 2. Of the remaining four $^{14}CO_2$ flux terms described in § 2.2, only the two disequilibrium terms were optimized during the inversion, while the nuclear and cosmogenic terms were held fixed.





## 3.4 Transport errors

A limitation of any OSSE making use of the same atmospheric transport model both to create and then to assimilate the observations is the implicit assumption that the transport is perfectly known, with the result that random and systematic transport errors cannot be adequately accounted for (e.g., Nassar et al. (2014); Liu et al. (2014); Hungershoefer et al. (2010); Chevallier et al. (2009)). This is true even when the elements of the model-data mismatch matrix, $R$, includes some prior estimate of the expected transport uncertainty. A more comprehensive way to estimate the impact of transport model error is to use different transport models for the simulation and assimilation steps (Chevallier et al., 2010b). This is, however, a non-trivial task since expertise to run multiple global transport models does not typically exist within a single research group.

In the absence of two entirely different transport models for the simulation and assimilation steps, in one of our experiments we make use of TM5 EI to assimilate synthetic data simulated using TM5 EIC. As described in § 2.1.1, these two model variants differ substantially in their representations of vertical transport, which is an especially important component of the atmospheric transport with regard to flux estimation, since vertical transport directly influences the residence time of air within the continental boundary layer (CBL) and therefore the relationship between tracer flux and simulated concentrations in the CBL.

To illustrate this for our case, Figure 3 shows the mismatch between modeled and measured vertical profiles of $SF_6$ over the continental US (Sweeney et al., 2015) for both TM5 EI and EIC. We once again consider $SF_6$ because it is a nearly inert gas (lifetime ~2000 years) and, like $CO_2^{ff}$, it has purely continental sources linked to industrial activity overwhelmingly in the northern mid-latitudes (http://edgar.jrc.ec.europa.eu/part_SF6.php), but without a substantial seasonal cycle (Miller et al., 2012)). Thus, we expect the vertical gradient of $SF_6$ over the continental US to depend on the strength of vertical mixing between the boundary layer and the free troposphere, and any systematic differences between simulated and observed gradients to provide an observational constraint on the representation of vertical transport processes in the different models. As shown in Figure 3, both models display a mean offset from observations of ~0.04 ppt in the free troposphere, even at Trinidad Head (THD), which is upwind of the continent. This uniform free tropospheric offset is therefore likely due to incorrect $SF_6$ emissions in Asia. Apart from the upper level offset, the $SF_6$ gradient of TM5 EIC is consistently and significantly closer to the observations, suggesting that the EIC vertical transport scheme better represents the real atmosphere. Moreover, the vertical gradient of $SF_6$ between 850 hPa and 400 hPa (i.e., between ~1.5 km and ~7.1 km above sea level) for the two models differs by an average of 0.025 ppt across all sites, which is larger than the 0.018 ppt $1\sigma$ spread across sixteen modern global transport models over mid-latitude continents found by Patra et al. (2011). This suggests that TM5 EI and TM5 EIC provide substantially different realizations of the transport not just at hemispheric scale (Figure 2), but also at a location and scale most relevant to an "imperfect transport" OSSE for the conterminous US (Figure 3).

## 3.5 OSSE Evaluation

Flux inversion OSSEs are often evaluated according to the so-called "uncertainty reduction", defined as the fractional reduction between the prior and posterior flux uncertainty (e.g., Rayner and O'Brien (2001); Hungershoefer et al. (2010)). That metric,





however, depends heavily on the prior uncertainties prescribed, and a large uncertainty reduction could easily arise from insufficient weighting of the prior during flux estimation. Moreover, iterative schemes such as the variational scheme used in TM5 4DVAR cannot estimate the full-rank posterior error covariance matrix. As detailed by Meirink et al. (2008) and Basu et al. (2013), the posterior covariance matrix $\hat{B}$ is calculated from the prior matrix $B$ in TM5 4DVAR as

$$\hat{B} = B + \sum_{i=1}^{i=n} \left( \frac{1}{\lambda_i} - 1 \right) (Lv_i)(Lv_i)^T \tag{5}$$

where $L$ is the "square root" of $B = LL^T$, and $\{\lambda_i, v_i\}$ are the $n$ leading eigenvalues and eigenvectors of the dimensionless Hessian

$$\mathcal{H} = I + L^T H^T R^{-1} H L \tag{6}$$

$H$ and $R$ being the same as in equation (3). In equation (5) $n$ denotes the number of iterations performed during the 4DVAR optimization, which is 40 in this study. In the limit of $n = n_{\text{state}}$, where $n_{\text{state}}$ is the dimension of $\mathbf{x}$ being estimated, eq (5) yields an approximation to the analytical posterior covariance matrix that over-estimates the error in posterior fluxes (Meirink et al., 2008; Bousserez et al., 2015). Because of these constraints, we focus here instead on the mismatch between the inversion-estimated fluxes and "true" fluxes. Since prior and true fluxes differ significantly in both space and time the ability of our inversion to recover the truth should serve as a rigorous test of our observational and inversion framework.

For the "perfect transport" OSSEs, we also evaluate the posterior correlation between $F_{\text{fos}}$ and $F_{\text{bio}}$ to assess the degree to which these fluxes can be retrieved independently using (a) $CO_2$ data only and (b) using $^{14}CO_2$ and $CO_2$ data together. Conventional $CO_2$-only inversions solve eq (1a), but $F_{\text{fos}}$ is prescribed and not optimized. However, if we were to solve eq (1a) for both $F_{\text{bio}} + F_{\text{oce}}$ and $F_{\text{fos}}$, in a $CO_2$-only system we would expect a large negative correlation between the "natural" and fossil fuel fluxes since under most circumstances $CO_2$ observations constrain the total flux and not its components. The magnitude of this correlation would be limited in part by how well the $CO_2$ observations constrained the total $CO_2$ budget for the domain of interest; in the limiting case of a perfectly constrained total $CO_2$ budget, this correlation would be -1. Assimilating $^{14}CO_2$ observations in order to solve both equations (1a) and (1c) simultaneously, we should expect a reduction in the magnitude of negative correlation due to the independent information $^{14}CO_2$ provides about $F_{\text{fos}}$. The amount of reduction in the correlation between $F_{\text{bio}} + F_{\text{oce}}$ and $F_{\text{fos}}$ thus serves as an objective metric of the ability of $^{14}CO_2$ observations to separate "natural" and fossil fuel $CO_2$ fluxes within our observational framework. In the case of the conterminous US, the "natural" $CO_2$ flux is largely equivalent to $F_{\text{bio}}$, or NEE.

The evaluation of this posterior flux correlation, however, is imprecise in a variational approach because $F_{\text{fos}} : F_{\text{bio}}$ correlations are derived from the approximate posterior covariance matrix $\hat{B}$ of eq (5) with $n \ll n_{\text{state}}$. To obtain a more accurate estimate of the posterior covariance (and hence correlation) matrix, we follow the prescription of Chevallier et al. (2007). The posterior covariance between any two elements $x_i$ and $x_j$ of the state vector $\mathbf{x}$ being estimated is

$$C_{ij}^{\text{apos}} = \langle (x_i^{\text{apos}} - \bar{x}_i^{\text{apos}})(x_j^{\text{apos}} - \bar{x}_j^{\text{apos}}) \rangle$$



where the ensemble average is taken over an ensemble of variational inversions, each of which starts from a different prior and assimilates a different set of measurements, such that the probability distribution of all the priors follows the prior covariance matrix $B$ of equation (3), and the probability distribution of all the measurements follows the model data mismatch (covariance) matrix $R$ of equation (3).

5      Choosing the number of inversions in the ensemble is a balancing act between statistical robustness and computer resources. Bousserez et al. (2015) recommended at least 50 inversions to estimate the posterior covariance matrix to within 10%. A key assumption in their recommendation was that the mean of the posterior estimates $\mathbf{x}^{\mathrm{apos}}$ corresponded to the analytical solution, i.e., each individual inversion had already "reached convergence" to within the analytical posterior error. In our case, due to the limited number of iterations performed (40 out of the theoretically required $n_{\mathrm{state}} = 4,095,000$), we cannot be sure that within 10 the ensemble, the $\mathbf{x}^{\mathrm{apos}}$ estimates are distributed with the analytical solution as their mean. However, in the case of our OSSEs, we know the analytical solution, which is the "true" flux of § 3.1. Therefore, for evaluating the posterior covariance between $F_{\mathrm{fos}}$ and $F_{\mathrm{bio}}$, we perform an ensemble of inversions where the prior fluxes are perturbations from the "true" flux following the statistics of $B$. This approach of perturbing around a known truth to better estimate the posterior covariance is similar to that used by, e.g., Liu et al. (2014). To be on the safe side of the recommendation of Bousserez et al. (2015), our ensembles contain 15 100 inversions each.

     Performing 100 independent inversions is computationally expensive. Therefore, we only evaluate the posterior correlation between $CO_2^{\mathrm{ff}}$ and $CO_2^{\mathrm{nat}}$ for two scenarios, (a) the "NRC 5000" scenario, and (b) the "NRC 5000" scenario without $^{14}CO_2$ observations. In an ideal system, for scenario (b) we expect to see large negative correlations between the posterior "natural" and fossil fuel $CO_2$ flux, at least over large areas where the total $CO_2$ flux is well constrained, and in scenario (a) we expect the 20 negative correlations to be measurably smaller.

## 4    Results

OSSE results are considered at scales ranging from monthly national totals, monthly totals for regions specified in Figure 5, and for groups of neighboring regions. Figures 6 and 7 compare monthly totals of the estimated fossil fuel $CO_2$ flux to specified "true" fluxes used to create the observations and the prior fluxes used in the inversions, for both "2010" and "NRC 5000" 25 measurement coverage. At the national scale, the monthly fossil fuel flux over the contiguous United States is recovered to within 5% (orange shaded region in Figure 6) for all but one month for the 2010 measurement coverage, while the national, annual total is recovered to better than 1% ("true" flux = 1497.5 TgC, estimated flux = 1497.2 TgC). For the considerably denser measurement coverage of NRC 5000, the monthly US fossil fuel flux is recovered to within 5% (and usually to within 3%) for all months, while the national, annual total is again recovered to better than 1% ("true" flux = 1497.5 TgC, estimated 30 flux = 1506.5 TgC). The impact of the increased coverage is more obvious when we consider smaller regions. Over the Eastern and Central US, the NRC 5000 scenario always yields monthly flux estimates that are within 5% of the "truth", and over the Central US the phasing of the NRC 5000 estimate is much closer to the "truth" than that for the 2010 coverage. Estimates for the Western US frequently deviate by more than 5% from truth, even for the NRC 5000 scenario. This is likely due to the



combination of the relatively small regional emissions and the difficulty of representing the transport over complex terrain. Even for our case of effectively "perfect transport", the elements of the transport that carry emissions from upwind regions to the sampling sites may be biased; indeed it appears that both 2010 and NRC 5000 observation networks are detecting a transported signal from a region with a larger emission signal and greater seasonality than the Western US (compared to

the "truth"). And, unlike other US regions, the Western US tends to lack constraint from upwind observations (i.e., over the Pacific), which are relatively sparse in both measurement scenarios.

Over smaller regions (i.e. those of Figure 5), monthly flux estimates deviate more significantly from the "truth" under both coverage scenarios (Figure 7). This is expected, since the number and distribution of observations and the information content of the prior ultimately limit the spatiotemporal scale at which independent flux estimates can be reliably obtained. NRC 5000

monthly flux estimates are as good as or better than 2010 coverage estimates over almost all regions. Over regions 1, 4, 7 and 9, the NRC 5000 flux estimate is almost always within 5% of the "true" fluxes, whereas over regions 3, 5, 6 and 8 the NRC 5000 estimate sometimes falls outside the 5% interval, but is always within 10% of the "truth". Over region 2 (Mountain US), even though the NRC 5000 flux estimate does not follow the "truth" closely (likely for reasons discussed with respect to the Western US above), it is closer to the "truth" on average than the 2010 coverage estimate. By contrast, the 2010 coverage estimate

consistently falls within the 5% error range only over region 9 (South Atlantic US), whereas over several regions (e.g., 3, 6, 7 and 8) its performance is significantly worse than the NRC 5000 estimate. The good performance of the 2010 coverage over the Southern Atlantic states, compared to other regions, may be due to the presence of a surface (tower) sampling site at Beech Island, SC (SCT) and aircraft profiles and surface measurements at Cape May, NJ (CMA), which are typically downwind of that region.

Figure 8 shows the accuracy of estimated annual total fossil fuel fluxes over the United States and several sub-regions. For all the regions, the prior annual emission estimate is outside a 5% margin around the "true" emissions (orange rectangles). For the relatively sparse 2010 coverage scenario, the "true" fluxes are recovered to within 5% for the US, the Eastern US, the Central US, and two out of the nine regions of Figure 5. Under the augmented NRC 5000 coverage scenario, annual total fossil fuel flux estimates are within 5% of the truth for the conterminous US and all of its sub-regions except one (Mountain US).

## 4.1   Correlation between $F_{\text{fos}}$ and $F_{\text{bio}}$ with and without $^{14}CO_2$ observations

Over large land areas, $CO_2$ observations constrain only the sum of biospheric and fossil fuel $CO_2$ fluxes, thus any attempt to separately estimate the two based on $CO_2$ observations alone should lead to large negative correlations between the two flux types. Any independent information on fossil fuel fluxes from $^{14}CO_2$ observations can be expected to result in a reduction in this negative correlation. To evaluate this, we calculate the posterior correlation between fossil fuel and biospheric fluxes for

two scenarios, (a) an inversion using only $CO_2$ data to estimate both fossil fuel and biospheric $CO_2$ fluxes, and (b) an inversion using both $CO_2$ and $^{14}CO_2$ data for the same purpose. The synthetic data sets in both cases are drawn from the "NRC 5000" coverage scenario. The method used to calculate the posterior correlation matrix was outlined in § 3.5. If $y^{\text{ff}}$ and $y^{\text{nat}}$ denote





the fossil fuel and "natural" CO2 flux aggregates over some spatiotemporal extent (e.g., North America over 2010), then the correlation between fossil fuel and "natural" fluxes over that extent is

$$r = \frac{\sum_{i=1}^{N} \left(y_i^{\mathrm{ff}} - \langle y^{\mathrm{ff}} \rangle\right) \left(y_i^{\mathrm{nat}} - \langle y^{\mathrm{nat}} \rangle\right)}{\sqrt{\sum_{i=1}^{N} \left(y_i^{\mathrm{ff}} - \langle y^{\mathrm{ff}} \rangle\right)^2} \sqrt{\sum_{i=1}^{N} \left(y_i^{\mathrm{nat}} - \langle y^{\mathrm{nat}} \rangle\right)^2}} \tag{7}$$

where $y_i$ is the estimate of the spatiotemporal flux aggregate from the $i^{\mathrm{th}}$ inversion, and $\langle y \rangle$ is the mean $y$ across all $N$ inversions.

Characterizing an error for $r$ is not straight-forward since $r$ is bounded within $\pm 1$ and does not have a normal distribution. We therefore estimate a confidence interval of $r$ using a bootstrap method (Efron and Tibshirani, 1994) in which we randomly resample the 100 inversions with replacement and calculate the correlation coefficient from that random drawing. We repeat this 50,000 times to produce a distribution of $r$. We report the median value of $r$, and call the range between percentiles 2.5 and 97.5 the "error" in $r$ (i.e., covering 95% of the values, analogous to $\pm 2\sigma$ limits in a normal distribution).

The median value of the posterior correlation $r$ and its error range (95% confidence interval) is evaluated for the "NRC 5000" scenario with and without $^{14}CO_2$ observations, and plotted in Figure 9 for the conterminous US and several sub-regions. For the inversion with only $CO_2$ data, we expect the correlation to be strongly negative over regions for which the total carbon budget is well constrained by the $CO_2$ observations. In Figure 9 this is seen, for example, for the conterminous US (called "United States") due to the strong observational constraint posed by the large number of $CO_2$ observations (37,884 for the year 2010 in the "NRC 5000" coverage). Results for the Eastern US also show a strong negative correlation because of the dense coverage in the "NRC 5000" network (Figure 4) for that area compared to the Central and Western US. The observational constraint on the total $CO_2$ budget is less stringent, and hence the negative correlation weaker, over smaller regions (such as the NY-NJ-PA tri-state area or the New England states) or for regions for which the "upwind" influence is less well characterized and the "downwind" area is not well defined (such as the Pacific coast and the Western US).

Over all regions in Figure 9 the addition of $^{14}CO_2$ data weakens the negative correlation between fossil fuel and biospheric $CO_2$ flux, indicating that $^{14}CO_2$ provides information needed to partition $CO_2$ flux components . Over all the large regions, this reduction is significant; the $95^{\mathrm{th}}$ percentile error bars barely overlap for the Central US, and for the Eastern, Western and conterminous US the error bars are well separated. These represent areas where fossil fuel and biospheric flux estimates can be separated based on $CO_2$ and $^{14}CO_2$ observations from the NRC 5000 network.

## 4.2 Carry-over bias in NEE

As discussed in § 1, errors in fossil fuel fluxes specified in traditional $CO_2$-only inversions (usually with zero prior uncertainty) may be expected to result in spatial and temporal biases in estimated NEE, which we refer to as carry-over bias. To evaluate the magnitude of potential carry-over bias, and the extent to which it may be reduced by assimilating $^{14}CO_2$ observations, we compare two inversions in which the prior fossil fuel $CO_2$ flux fields are deliberately biased. The first is the NRC 5000 dual $CO_2 + {}^{14}CO_2$ inversion already discussed. The second is a $CO_2$-only inversion in which we estimate biospheric and oceanic



fluxes of $CO_2$ by assimilating synthetic $CO_2$ observations from the NRC 5000 network, but not $^{14}CO_2$ observations. For both inversions, the prior fossil fuel flux is from EDGARv4.2 FT2010 and the prior biospheric flux is from SiB CASA, as described in § 3.3. As can be seen in Figures 6, 7 and 8, both the annual and monthly totals for the prior fossil fuel fluxes differ markedly from the true fossil fuel fluxes for the US and all sub-regions. Using the entire conterminous US as an example, and assuming

stringent total carbon constraint based on the large number of $CO_2$ observations in the NRC 5000 scenario, we may anticipate monthly carry-over biases as large as 100-200 TgC/yr based on differences between true and prior fluxes in winter and mid-summer (e.g., 185 TgC/yr in January 2010, 133 TgC/yr in July 2010, and 176 TgC/yr in December 2010).

Estimated biospheric fluxes for the two inversions are given along with "true" and prior biospheric fluxes as both monthly and annual net totals in Figure 10 for the conterminous US and several sub-regions. In all cases, both inversion estimates (those

with and without $^{14}CO_2$ observations) migrate away from the specified prior biospheric fluxes and lie close to "true" biospheric fluxes. This is due to the observational constraints provided by the very large number of synthetic $CO_2$ measurements and the fact that even the largest potential carry-over bias (e.g., 188 TgC/yr in February 2010 for the US) is small relative to either prior or "true" monthly NEE, which is typically at least an order of magnitude larger. However, we note that for regions that are rich in both $CO_2$ and $^{14}CO_2$ observations, such as the Eastern US, we resolve differences between the cases with and

without $^{14}CO_2$ assimilation that are directly comparable to differences in the underlying fossil fuel inventories. These results indicate that carry-over biases that would otherwise go unresolved can be in part overcome by adding observational constraints from $^{14}CO_2$. For example, the fossil fuel prior in February 2010 over the Eastern US is biased low by 154 TgC/yr, which results in an NEE estimate 153 TgC/yr higher than the "truth" if $^{14}CO_2$ data are not assimilated, but only 78 TgC/yr higher than the "truth" if $^{14}CO_2$ data are assimilated. Similarly, in December 2010, the fossil fuel prior over the Eastern US is biased

low by 163 TgC/yr, resulting in a bias in the estimated NEE of 133 TgC/yr without assimilation of $^{14}CO_2$ and only 9 TgC/yr with $^{14}CO_2$ observations. These results indicate that carry-over biases that would otherwise go unresolved can in large part be overcome by adding observational constraints from $^{14}CO_2$.

For the three US sub-regions in Figure 10 (right panel), the annual NEE estimate with $^{14}CO_2$ is closer to "truth" than without. However, the reverse is true for annual NEE aggregated over the conterminous US (i.e. the sum of the three sub-regions). This

is due to a cancellation between the Western US (where the $CO_2$-only NEE estimate is too negative) and the other two regions (where the $CO_2$-only estimate is too positive compared to the "truth").

### 4.3  Imperfect transport OSSE

As mentioned in § 3.4, we performed an inversion with intentionally biased transport. That is, we simulated $CO_2$ and $^{14}CO_2$ measurements with "true" fluxes in TM5 EIC, and assimilated those observations using TM5 EI. As noted in section 3.4,

forward simulations of an inert tracer sourced largely from the northern continents ($SF_6$, which is in this respect similar to fossil fuel $CO_2$) produce substantially different vertical profiles over the conterminous US for the two model versions (Figure 3), indicating that the two models represent meaningfully different realizations of atmospheric transport.

Figure 11 shows the monthly fossil fuel fluxes estimated over the Unites States and three of its sub-regions for both biased transport (NRC 5000 (EI)) and what is effectively "perfect transport" (NRC 5000). For assimilation of observations using TM5





EI, the monthly flux estimates over the conterminous United States (and over its three large-scale sub-regions) no longer lie within 5% of the "true" fluxes. The flux estimates with biased transport are in this case uniformly low, consistent with our understanding of the primary difference between EI and EIC transport schemes involving vertical entrainment and detrainment fluxes over the northern temperate latitudes. As seen for forward simulations in Figure 3, EIC tends to better ventilate the CBL

such that the surface signal is more efficiently transferred to the well mixed free troposphere compared to EI, which allows more signal to build up within the CBL. Thus, TM5 EI requires smaller surface fluxes in order to recover the surface layer signal simulated by TM5 EIC; annual fossil fuel flux estimates from EI transport are thus in all cases lower than the estimates from EIC transport (Figure 8).

As outlined in § 2.1.1 and § 3.4 TM5 EI and TM5 EIC differ significantly in terms of their respective vertical transport

schemes, giving rise to large differences in transported tracer distributions at the global scale and, importantly, over the northern mid-latitude continents. TM5 EI is in particular demonstrably biased compared to the ensemble of transport models used in most state-of-the-art global inversions according to several metrics considered by Patra et al. (2011). Thus, while the differences between our fossil fuel $CO_2$ flux estimates serve as a demonstration of the potential biases that can arise from poor or differing representations of the real transport, they almost certainly exaggerate flux biases likely to be seen amongst models that are well

validated against observations. Conversely, our results with effectively "perfect transport" serve to demonstrate that assimilation of $^{14}CO_2$ along with $CO_2$ observations has the potential to yield direct, independent "top down" observational constraints on fossil fuel emission at sub-continental, regional scales ($\sim$250,000 km$^2$) with uncertainties comparable to those estimated for "bottom up" inventories. Ongoing improvements in tracer transport models along with rigorous evaluation of transported tracer distributions against a growing network of observations, of the kind we show for $SF_6$ in Figures 2 and 3, provide a clear path

towards a more complete realization of the full potential of the dual $^{14}CO_2$ and $CO_2$ assimilation capability described in this work.

## 5   Conclusions

In this work we develop and present a new dual tracer inversion framework that makes use of the present and anticipated networks of precise atmospheric $^{14}CO_2$ measurements to simultaneously estimate fossil fuel derived and biospheric fluxes of

$CO_2$. Using a set of Observation System Simulation Experiments (OSSEs), we demonstrate the ability of atmospheric $CO_2$ and $^{14}CO_2$ measurements to recover previously specified "true" fossil fuel $CO_2$ emissions over North America. As expected, the accuracy of the flux estimates depends both on the coverage of the measurement network and the spatiotemporal scale of analysis. We simulated two coverage scenarios, namely the coverage of the NOAA GGRN network in 2010 (969 $^{14}CO_2$ measurements over North America), along with an augmented coverage of $\sim$5000 $^{14}CO_2$ measurements over North America

("NRC5000"), as recently recommended by the US NAS (Pacala et al., 2010). With the 2010 coverage, we recover "true" annual total fossil fuel emissions over the conterminous US to better than 1% and over several highly emissive sub-regions to within 5%. For "NRC 5000" coverage, we also recover monthly emissions to within 5% for the United States. For all but one of nine sub-regions, we also recover the monthly emission to within 5% for at least nine months of the year with the "NRC





5000" coverage (where, for the sub-region which is the exception, emissions are small and upwind observations are sparse). For regions with a good constraint on the total $CO_2$ flux based on large numbers of $CO_2$ observations in the NRC 5000 scenario, the anticipated $^{14}CO_2$ coverage allows for detection of and substantial reduction in biases in regional NEE that would otherwise arise from erroneous specification of the fixed fossil fuel $CO_2$ emission in a traditional $CO_2$-only inversion. Additionally, we

evaluate biases in fossil fuel $CO_2$ flux estimates that can arise from poor representation of atmospheric transport and show how the growing network of other tracer measurements may be used to select and improve the best transport models. For the best models, our ability to recover fossil fuel emissions over the US should approach that of our idealized OSSEs and be comparable to that for most "bottom up" fossil fuel emission inventories with estimated annual and monthly regional uncertainties of 5-10%. In a future world with anticipated national commitments to reduce $CO_2$ emissions (e.g. Intended Nationally Determined

Contributions, or INDCs, http://unfccc.int/focus/indc_portal/items/8766.php), such a capability could provide for independent "top down" verification of such commitments for the US and other areas where atmospheric observing networks are or can be established.

*Acknowledgements.* We would like to thank Colm Sweeney for providing aircraft-based measurements of $SF_6$ and Ed Dlugokencky for marine boundary layer measurements of $SF_6$, Colin Lindsay for homogenizing global neutron monitor data and oceanic $^{14}CO_2$ data from

multiple sources, and Nicolas Bousserez for useful discussions on uncertainty quantification. All computations for this work were performed on the Zeus and Theia clusters of the NOAA Research & Development High Performance Computing System (RDHPCS).





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



**Table 1.** Inversions and model runs performed in this work.

| Section (§) Figure (F) | Experiment | Model run | Obs network | Transport | Initial fluxes | Optimized fluxes[a] |
|---|---|---|---|---|---|---|
| § 3.1, § 3.2 | 2010 (simul) | forward | 2010 coverage | TM5 EIC | "true" | None |
| § 3.1, § 3.2 | NRC 5000 (simul) | forward | NRC 5000 | TM5 EIC | "true" | None |
| § 4, F6, F7, F8 | 2010 (assim) | inverse | 2010 coverage | TM5 EIC | "prior" | all except $F_{nuc}$, $F_{cosmo}$ |
| § 4, F6, F7, F8, F11 | NRC 5000 (assim) | inverse | NRC 5000 | TM5 EIC | "prior" | all except $F_{nuc}$, $F_{cosmo}$ |
| § 4.3, F8, F11 | NRC 5000 (EI) | inverse | NRC 5000 | TM5 EI | "prior" | all except $F_{nuc}$, $F_{cosmo}$ |
| § 4.1, F9 | NRC 5000 | inverse | NRC 5000 | TM5 EIC | "truth" + random perturbations | all except $F_{nuc}$, $F_{cosmo}$ |
| § 4.1, F9 | NRC 5000 (no $^{14}CO_2$) | inverse | NRC 5000 (only $CO_2$) | TM5 EIC | "truth" + random perturbations | all except $F_{nuc}$, $F_{cosmo}$ |
| § 4.2, F10 | NRC 5000 (traditional) | inverse | NRC 5000 (only $CO_2$) | TM5 EIC | "prior" | only $F_{bio}$, $F_{oce}$ |

[a] Of the seven flux terms in equations (1)





**Table 2.** Spatial covariance parameters of equation 4 for different categories.

| Category | Optimized | $C_\mathbf{r}$ type | L (km) | $\sigma$ |
|---|---|---|---|---|
| $F_{bio}$ | yes | **e** | 200 | $0.5 \times$ respiration |
| $F_{oce}$ | yes | **e** | 1000 | $1.57 \times$ absolute flux |
| $F_{fos}$ | yes | **h** | 500 | $2.5 \times$ inter-prior spread[a] |
| $F_{ocedis}$ | yes | **r** | – | $0.2 \times$ absolute flux[b] |
| $F_{biodis}$ | yes | **r** | – | $0.5 \times$ absolute flux[c] |
| $F_{nuc}$ | no | – | – | – |
| $F_{cosmo}$ | no | – | – | – |

[a] For fossil fuel $CO_2$ flux, the "inter-prior spread" denotes the spread between three fossil fuel inventories, CarbonTracker/Miller, Carbon-Tracker/Vulcan and ODIAC (Oda and Maksyutov, 2011). For defining the region boundaries across which the prior flux correlation goes to zero, we used nine divisions of the continental Uinted States defined by the US Census Division (www.eia.gov/forecasts/aeo/pdf/f1.pdf), shaded in Figure 5. The rest of North America falls into a single region, while other continents, namely South America, Europe, Africa, Asia and Australia form five separate regions. All ocean pixels fall in one single region, while non-optimized pixels (Greenland and Antarctica) fall into one region.

[b] The world's oceans are divided into the eleven TRANSCOM ocean regions (Law et al., 2000).

[c] Outside North America, the land is divided up into nine TRANSCOM land regions. Inside North America, the North American temperate region is by itself, while the North American boreal region is further subdivided into 11 regions used by CarbonTracker 2013b (http://www.esrl.noaa.gov/gmd/ccgg/carbontracker/CT2013B_doc.php, § 8.1.1).

**Table 3.** Sampling frequency of $CO_2$ and $^{14}CO_2$ measurements at the sites of our hypothetical "NRC 5000" network. Even though Figure 4 only displays sites over the conterminous US and part of Canada, our NRC 5000 network also has some background sites such as South Pole and Mauna Loa. The numbers below include all sites, globally.

| Site type | # of sites for $CO_2$ | Sampling freq. for $CO_2$ | # of sites for $^{14}CO_2$ | Sampling freq. for $^{14}CO_2$ |
|---|---|---|---|---|
| Tower | 62 | 2/day | 35 | 2/week |
| Flask | 76 | 1/week | 21 | 1/week |
| Aircraft | 19 | 1/week, up to 16 altitudes | 11 | 1/week, 3 altitudes |
| Cruise | 2 | 1 transect/month, every 5° latitude | – | – |





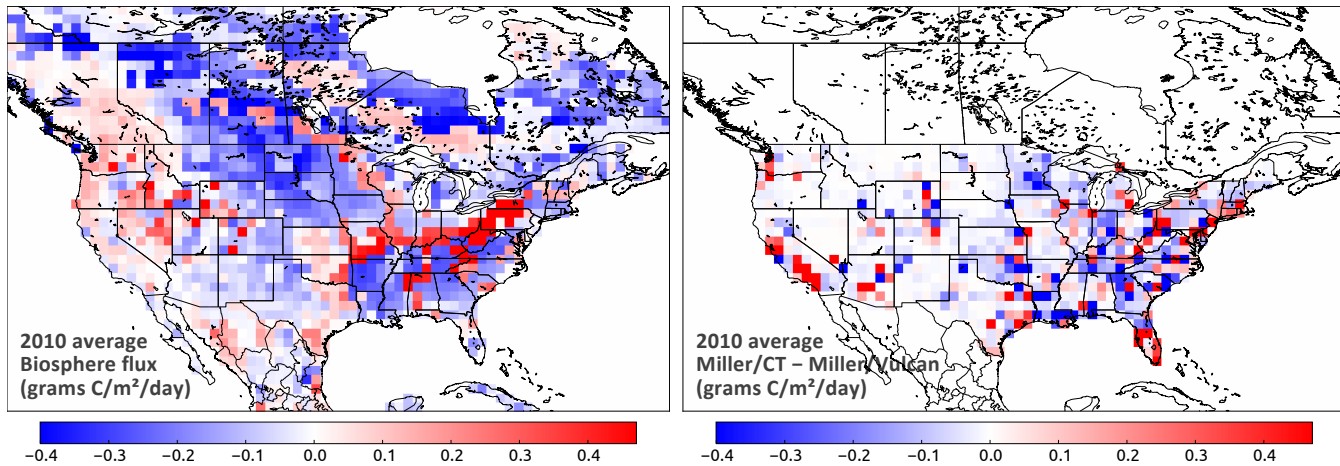

**Figure 1.** Comparative magnitudes of the annual average NEE estimated by CarbonTracker 2013B (left) and the difference between two fossil fuel inventories, Miller/CT and Miller/Vulcan (right). CarbonTracker (carbontracker.noaa.gov) is an atmospheric inversion which estimates $CO_2$ surface fluxes given atmospheric $CO_2$ measurements and "perfectly known" fossil fuel emissions. Miller/CT is the fossil fuel emission map prescribed in CarbonTracker 2013B, while Miller/Vulcan is a redistribution of the Miller/CT annual total fossil fuel $CO_2$ emission over the conterminous United States according to the spatiotemporal pattern of the Vulcan fossil fuel inventory (Gurney et al., 2009). While annual total emissions over the conterminous US for the two inventories are the same (i.e., the reds and blues in the right figure sum to zero), over individual $1° \times 1°$ grid cells their difference can be comparable to the NEE estimated at the same location.





**Figure 2.** The observed and modeled latitudinal gradients of $SF_6$, estimated as the difference between $SF_6$ concentration at marine boundary layer sites of the NOAA ESRL GGRN (http://www.esrl.noaa.gov/gmd/ccgg/ggrn.php) and the South Pole. Observations and models span ten years from 2002 to 2011. For each site, we account for time-dependent changes by calculating a linear trend from observed $SF_6$ and removing that from all three time series (observed, TM5 EI, TM5 EIC). All observations were binned by latitude in 5° increments and averaged. The bottom panel shows the number of samples averaged per latitude bin. The error bars denote $\pm 2\sigma$ intervals, where $\sigma$ is the standard error of the mean difference w.r.t. South Pole. $SF_6$ mole fractions used here are available on request.



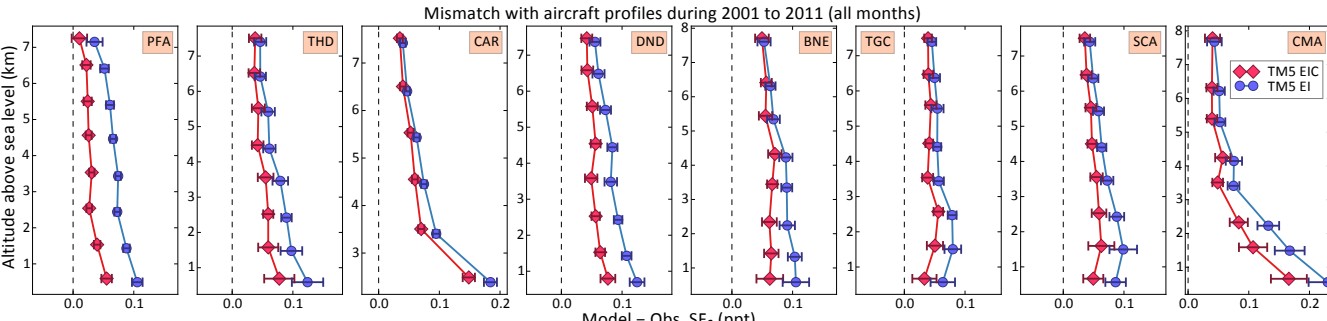

**Figure 3.** Average difference between observed and modeled aircraft profiles of $SF_6$ at eight different sites over the continental United States for the period 2001 to 2011 (inclusive). The error bars denote $\pm 2\sigma$ intervals, where $\sigma$ is the standard error of the mean difference between each model and observations. The locations of the profiles are identified by three-letter site codes. Details for each site can be found at http://www.esrl.noaa.gov/gmd/dv/site/site_table2.php. Atmospheric mole fractions of $SF_6$ were simulated using the EI and EIC variants of the TM5 transport model. Three-dimensional initial conditions on January 1, 2000 were based on (a) vertical gradients from the end (January 1, 2006) of a previous six year TM5 run (with initial conditions which included a specified latitude gradient but no vertical gradient) and (b) the January 1, 2000 smoothed marine boundary layer latitudinal gradient (Masarie and Tans, 1995) derived from $SF_6$ observations from the NOAA ESRL GGRN. The January 1, 2006 vertical gradients were zonally averaged, scaled back to January 1, 2000 and then added to the observed latitude gradient to create a zonally uniform but vertically and meridionally variable field. $SF_6$ emissions for the run were based on the spatial emission pattern from EDGAR v4.2 scaled to match the annual increases in $SF_6$ emissions derived from the observed $SF_6$ growth rate (assuming no atmospheric $SF_6$ loss). $SF_6$ mole fractions used here are available on request.





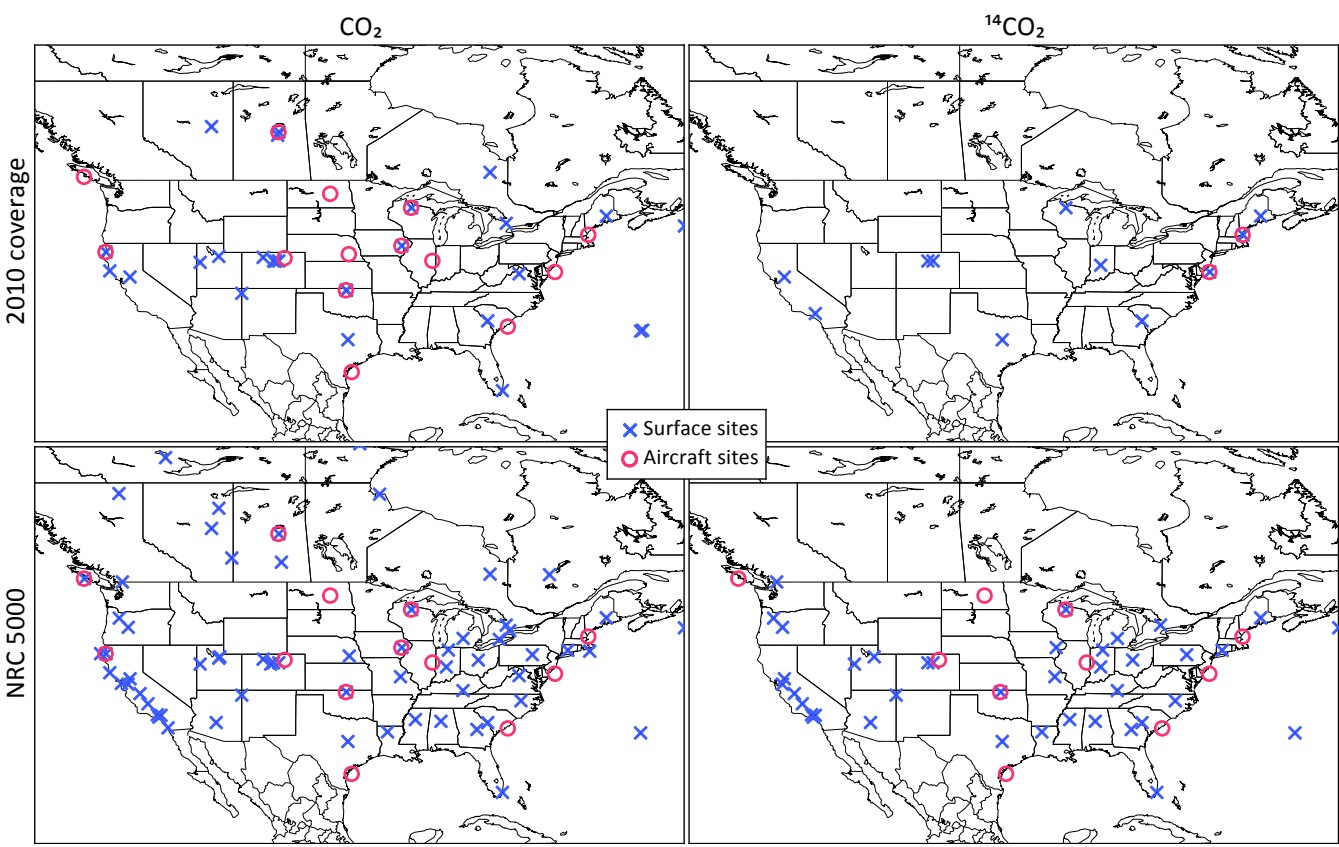

**Figure 4.** The sites for which $CO_2$ and $^{14}CO_2$ measurements were simulated and then assimilated in our OSSEs for two different coverage scenarios, "2010" and "NRC 5000", as described in the text.





**Figure 5.** Nine regions defined by the US Census Division, over which we aggregate our fossil fuel $CO_2$ flux estimates (www.eia.gov/forecasts/aeo/pdf/f1.pdf).





**Figure 6.** Monthly total emissions estimates for "2010" and "NRC 5000" network scenarios, along with prior and "true" fluxes, aggregated for the conterminous US and neighboring groups of regions identified in Figure 5. The orange band depicts the ±5% margin around the "true" fluxes, and the numbers next to region names refer to the region labels in Figure 5.





**Figure 7.** Monthly total emissions estimates for "2010" and "NRC 5000" network scenarios, along with prior and "true" fluxes for individual regions identified in Figure 5. The orange band depicts the ±5% margin around the "true" fluxes, and the numbers next to region names refer to the region labels in Figure 5.





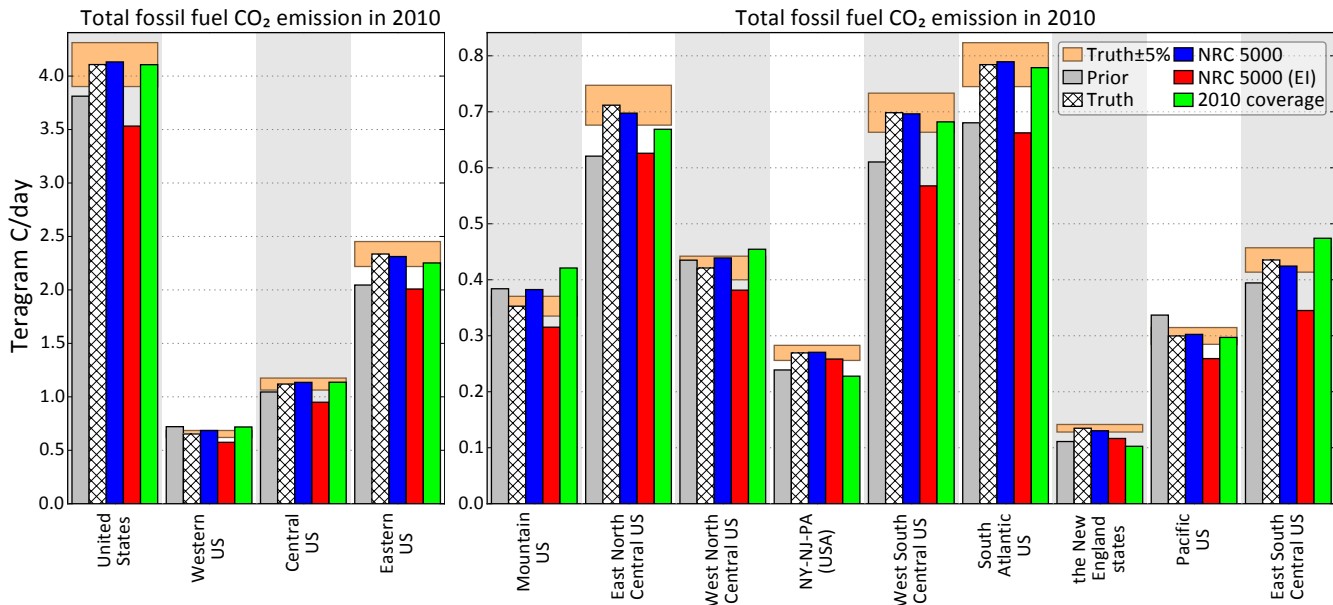

**Figure 8.** Annual total fossil fuel $CO_2$ emissions estimates for "2010" and "NRC 5000" network scenarios along with "true" and prior fluxes aggregated for the conterminous US, individual regions and neighboring groups of regions identified in Figure 5. The orange rectangles denote the ±5% range around the "true" emission each region.

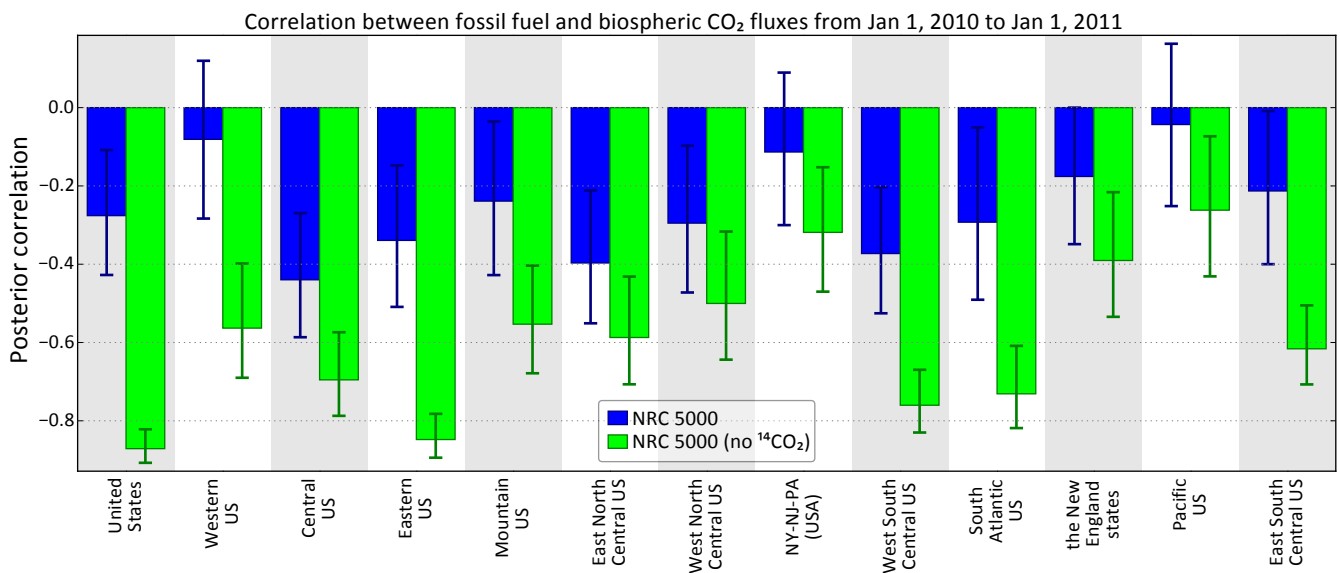

**Figure 9.** Posterior correlation between fossil fuel and biospheric $CO_2$ fluxes obtained with and without $^{14}CO_2$ measurements for the "NRC 5000" network scenario.



**Figure 10.** Monthly net biospheric $CO_2$ flux estimates for the "NRC 5000" network scenario with and without $^{14}CO_2$ observations along with prior and true fluxes aggregated for the conterminous and Eastern US (left) and annual net biospheric and fossil fuel fluxes for the conterminous US and groups of neighboring regions (right). As discussed in the text, the "NRC 5000 (traditional)" inversion does not optimize fossil fuel fluxes and does not assimilate $^{14}CO_2$ observations. For both the inversions above, large numbers of $CO_2$ observations in the NRC 5000 scenario drive the biosphere flux estimates toward "true" fluxes, while adding $^{14}CO_2$ helps to address carry over bias arising from erroneous specification of the fossil fuel prior.





**Figure 11.** Monthly total fossil fuel $CO_2$ emission estimates along with prior and true fluxes aggregated for the conterminous US and neighboring groups of regions identified in Figure 5, using "perfect" ("NRC 5000") and intentionally biased transport ("NRC 5000 (EI)"). As discussed in the text, estimates for biased transport are in this case uniformly low because of systematic differences in the vertical transport between the two model variants.