# Peer review of "Separation of biospheric and fossil fuel fluxes of $CO_2$ by atmospheric inversion of $CO_2$ and ${}^{14}CO_2$ measurements: Observation System Simulations"

_Atmospheric Chemistry and Physics, 2016_

## Referee Comment (RC1) · Anonymous Referee #1 · 16 Feb 2016

Basu and colleagues investigate the potential to quantify fossil fuel CO2 emissions from atmospheric CO2 and 14CO2 data. They use synthetic atmospheric inversions for North America assuming either a data availablity as existing in 2010 or a realistic assumed extension for the future. If this data availablity becomes true, Basu and colleagues conclude that monthly fossil fuel emissions can be quantified within 5% in the more densly observed regions of the USA, which can be used to independently verify reported emissions. This is a highly relevant piece of information, as the build-up of observational capacities is expensive and requires time.
[Figure]

The results are obtained by widely accepted methods which are convincingly applied and described. The presentation is very clear and covers the relevent aspects. I clearly recommend to publish this work in ACP.

I only have a few minor comments and suggestions:

p2 l6: maybe add "e.g., Andres..."

p3 eq1: Strictly, this equation is only true for a well-mixed volume, not for the whole "atmospheric burden". I'm aware that the message of the equation is about the individual flux components, but maybe remove the ambiguity by saying e.g. "local $CO_2$ mixing ratio" (or "local atmospheric mass balance" in line 15).

p4 l1 and beyond: It seems the unit is misspelled and should be "PgC/yr * per mil" (not "/ per mil").

p6 l13: It would be easier to use the labels "e", "r", and "h" already in the enumeration in lines 8-12.

p9 l5: Clearify if you sampled at the two times *each day* (I assume so but it remains open).

Sect 3.4: You denote the absence of transport error as a limitation, but I'd actually see this as an advantage, because the result specifically diagnoses the constraining power of the observations. (I nevertheless agree that the investigation of transport model errors as done later is interesting information.)

p10 l15: Fig 3 is referred to later than Fig 4.

p11 l4-12: As you do not use the explicit covariance matrices anyway, I feel this description rather confuses and could be omitted.

p11 l24: I agree the metric is objective, but how to interpret it quantitatively? (see comment below)

[Figure]

p12 l8: I'm astonished why you cannot be sure about your convergence. Couldn't plot the result as a function of iteration count and check if the behaviour is still transient?

Sect 4: In rare cases (e.g. fig 7 region 5 in summer), the 2010 stations give considerably better fit - do you know why this is?

p13 l1-4: You invoke transport model errors (which is somehow contradicted by the absence of model errors), but couldn't that just be "leakage" from neighbouring regions due to incomplete seperability?

p14 l13: To make this more understandable, say what you in contrast expect for less well constrained regions, and why.

p14 l21-25: I'm not sure I fully understand this. Wouldn't the criterion for separability be a correlation range overlapping zero? As said earlier, I'm not fully convinced that the correlation coefficient can be interpreted quantitatively.

Conclusion: Ingeborg Levin and colleagues had concluded that fossil fuel emission changes can be detected from 14CO2 data if larger than 7–26% for five-year averages, being limited also due to interannual variations. This seems somewhat more pessimistic than your results. Can you add a comment whether (or to which extend) these results are compatible, and why? (reference: I. Levin et al, Naturwissenschaften (2008) 95:203–208, DOI 10.1007/s00114-007-0313-4)

---

## Referee Comment (RC2) · F.R. Vogel (Referee) · 24 Feb 2016

The study presented here by Basu et al. investigates the potential of an inversion framework when assimilating 14CO2 and CO2 using an Observation System Simulation. The authors describe their pseudo-data experiment thoroughly and concisely. The design parameters are driven by previous suggestions to expand the 14C observing network to improve the capability to estimate annual total fossil fuel CO2 emissions from the contiguous United States. The assumptions for the modelling framework seem all very well-founded in experience and previous research in this field. Although not all

aspects of errors and error correlations are discussed in detail this study is definitely a clear step forward. The topic i.e. using independent measurements to assess reported national GHG emissions is timely and presented in a very clear way. Both the quality and topic are well suited for ACP and I fully recommend publication, as is.

Minor comments: Page 3 equation (1b) Please consider that the mass-balance for 14CO2 is only valid for d14C not D14C. The author discuss the issue of d13C corrections impacting D14C this confusion ca be avoided putting the mass balance for 14CO2 and then mentioning the assumptions made to arrive at a mass balance for D14C. e.g. https://journals.uair.arizona.edu/index.php/radiocarbon/article/downloadSuppFile/16347/212 The impact of the approximation in (1b) seems negligible.

Page 16/17 conclusions The authors briefly discuss the potential impacts of model transport errors (investigated in section 4.3) and the added value of measurements of auxiliary species. Would you be able to advise on how much more model improvement is needed i.e. should this be an equally/less/more important part of developing the suggested future emission monitoring system?

---

## Author Comment (AC1) · 6 Mar 2016

**Response to reviewer 1**

Sourish Basu, John B. Miller, Scott Lehman

March 6, 2016

We thank the reviewer for a prompt and favorable review, and for the helpful suggestions they made. Please find below our responses to the specific comments below. The reviewer's comments are in *blue italics*, our responses are in normal text. Portions of a response that reflect changes in the main text are in "double quoted red".

*p2 l6: maybe add "e.g., Andres: : :"*

Added "cf. Andres et al"

*p3 eq1: Strictly, this equation is only true for a well-mixed volume, not for the whole "atmospheric burden". I'm aware that the message of the equation is about the individual flux components, but maybe remove the ambiguity by saying e.g. "local CO2 mixing ratio" (or "local atmospheric mass balance" in line 15).*

If C refers to the total atmospheric budget (i.e., total number of moles) of $CO_2$ and $\Delta_{atm}$ refers to the average atmospheric signature of $^{14}CO_2$, as we intended, then equation (1) is valid even in a non-well mixed atmosphere. In fact, it's not valid locally because of transport. However, if C denotes the mixing ratio of $CO_2$, which is how we suspect the reviewer interpreted it, then it's correct that the mass balance of equation (1) is valid globally only for a well mixed volume. For $CO_2$ and $^{14}CO_2$, the atmosphere is well mixed over the time scale of a few years, and in the $CO_2$ literature the mass balance of equation (1) is considered valid even for one year.

*p4 l1 and beyond: It seems the unit is misspelled and should be "PgC/yr * per mil" (not "/ per mil").*

Good point. There was indeed a mistake in the units. All the relevant units in the text have been corrected to "PgC ‰/yr".

*p6 l13: It would be easier to use the labels "e", "r", and "h" already in the enumeration in lines 8-12.*

The item labels are now "e:", "r:" and "h:" instead of (a), (b) and (c).

*p9 l5: Clearify if you sampled at the two times \*each day\* (I assume so but it remains open).*

Yes, we sampled the "true" $CO_2$ fields twice each day at tower sites. The new (clarified) text reads "At tower sites, we sampled the "true" $CO_2$ field twice a day at the highest intake height, at 00:30 and 03:30 local solar time (LST) for mountaintop sites and at 12:30 and 15:30 LST otherwise."

*Sect 3.4: You denote the absence of transport error as a limitation, but I'd actually see this as an advantage, because the result specifically diagnoses the constraining power of the observations. (I nevertheless agree that the investigation of transport model errors as done later is interesting information.)*

If the purpose of an OSSE is to evaluate the constraining power of a set of observations, given the inevitable progress towards increasingly accurate transport models in the future, then indeed "perfect transport" is not a limitation. If the purpose of an OSSE is to answer the question "What could we do today if we had ~5,000 $^{14}CO_2$ observations over North America per year?", then assuming perfect transport is a limitation, since currently transport model error is often the Achilles' heel of top-down flux estimates. We thank the reviewer for pointing out this distinction. We have changed the opening few sentences of section 3.4 to:

"The OSSEs described above allow for an accurate assessment of our ability to calculate fossil and biosphere fluxes given different sets of $^{14}CO_2$ and $CO_2$ observations, in the limit of perfectly known atmospheric transport (note, however, that the elements of the model-data mismatch matrix R are inflated to account for expected transport uncertainty). The performance of an inversion of real $^{14}CO_2$ data will be limited not only by the observations ingested, but also by errors in simulated atmospheric transport not adequately represented by R (e.g., Nassar et al. (2014); Liu et al. (2014); Hungershoefer et al. (2010); Chevallier et al. (2009))."

*p10 l15: Fig 3 is referred to later than Fig 4.*

We have rearranged the figures so that the first reference to Figure 3 comes before that of Figure 4.

*p11 l4-12: As you do not use the explicit covariance matrices anyway, I feel this description rather confuses and could be omitted.*

We agree with the reviewer. We have omitted the description of the calculation of the covariance matrix in TM5 4DVAR, referring to earlier papers instead.

*p11 l24: I agree the metric is objective, but how to interpret it quantitatively? (see comment below)*

Please see our response to the reviewer's comment about page 14, lines 21-25.

*p12 l8: I'm astonished why you cannot be sure about your convergence. Couldn't plot the result as a function of iteration count and check if the behaviour is still transient?*

The issue here is the definition of "convergence". Since iterative schemes are not expected to reach the analytical solution, convergence is usually defined in a variety of ways independent of proximity to that solution. For example, convergence can be defined as the reduction of the cost function (or the norm of the gradient of the cost function) from its initial value by a certain factor. This has the disadvantage of being misleading if the prior cost function (or its gradient) is very high due to, e.g., bad prior fluxes. To counter this, some people define convergence as the absolute value of the norm of the gradient being lower than some pre-determined value, although that has the disadvantage that the pre-determined value is not obvious to specify for a given problem. Yet other people specify convergence as a fixed number of iterations (which is the approach in our study), based on past experience (which, for sure, has its own limitations). The convergence relevant for page 12, line 8 is defined as proximity to the analytical solution, as in the conclusions of Bousserez et al. (2015) are valid only if the posterior solutions are distributed around the analytical solution. To our knowledge, no one has yet devised a convergence criterion for a variational system which would guarantee a certain proximity to the "exact" solution. This is what motivated our statement of not being sure of how close our posterior was to the analytical solution.

The referee suggests plotting the result as a function of iteration count to check for the subsidence of transient behavior. This sounds reasonable, but is difficult to implement in practice because it depends on which result we consider. For example, Meirink et al. (2008) showed that fluxes over larger scales converge quicker than fluxes over smaller scales. As a function of the number of iterations, for example, the global annual total flux converges quickly, global monthly fluxes take somewhat longer, and regional fluxes take even longer. Therefore, arriving at a unique definition of convergence by looking at the transience of results is difficult.

*Sect 4: In rare cases (e.g. fig 7 region 5 in summer), the 2010 stations give considerably better fit - do you know why this is?*

We noticed this also. Since an inversion tries to fit all observations "on average", we suspect that the better fit of the 2010 coverage for some months and regions is at the expense of worse fit at other regions. In general, increasing observation coverage will always improve the fit to the true fluxes "on average", but there is no guarantee of monotonic improvement at all space and time scales.

*p13 l1-4: You invoke transport model errors (which is somehow contradicted by the absence of model errors), but couldn't that just be "leakage" from neighbouring regions due to incomplete seperability?*

The referee is correct that transport errors due to complex terrain cannot be the reason here. We have therefore removed that wording. The remaining text points precisely to the issue the referee is referring to, i.e., "leakage" from neighboring regions (in combination with the lack of "upwind" measurements).

*p14 l13: To make this more understandable, say what you in contrast expect for less well constrained regions, and why.*

We have changed the sentence to "For the inversion with only $CO_2$ data, we expect the correlation to be strongly negative (i.e., close to −1) over regions for which the total carbon budget is well constrained by the $CO_2$ observations, and less negative (i.e., closer to 0) over regions with fewer observational constraints." Later in the paragraph there are examples of both strongly and weakly constrained regional $CO_2$ budgets and the resultant correlations.

*p14 l21-25: I'm not sure I fully understand this. Wouldn't the criterion for separability be a correlation range overlapping zero? As said earlier, I'm not fully convinced that the correlation coefficient can be interpreted quantitatively.*

We agree with the referee that it is hard to quantitatively interpret the posterior correlation coefficient in general, since only specific values (such as zero and minus one) have strict physical meanings. However, even for intermediate values, the principle holds that $r$ values closer to zero are "better" for the separability between fossil fuel and biospheric fluxes. The point we want to make here is that the addition of $^{14}CO_2$ data always takes the correlation coefficient in the right direction, towards more separability, and in some cases significantly so, as measured by the non-overlap of the 95[th] percentile error bars.

*Conclusion: Ingeborg Levin and colleagues had concluded that fossil fuel emission changes can be detected from 14CO2 data if larger than 7-26% for five-year averages, being limited also due to interannual variations. This seems somewhat more pessimistic than your results. Can you add a comment whether (or to which extend) these results are compatible, and why? (reference: I. Levin et al, Naturwissenschaften (2008) 95:203-208, DOI 10.1007/s00114-007-0313-4)*

We thank the reviewer for pointing out this work. As far as we can see, our work is compatible and complementary to Levin and Rödenbeck (2007). They address the question of the percentage change

required in the enhancement of fossil fuel $CO_2$ ($\Delta FFCO_2$) to be detectable by $^{14}CO_2$ measurements at Schauinsland and Heidelberg (compared to the background site of Jungfraujoch). They conclude that a smaller percentage change is detectable from a high emitting area (vicinity of Heidelberg) compared to a low emitting area (vicinity of Schauinsland). This is consistent with what we find, e.g., even with the sparser 2010 network we can in most months estimate the fossil fuel $CO_2$ flux to within 5% for the high emitting Eastern US but not for the lower emitting Central and Western US. Beyond this similarity, however, it is difficult to compare numbers from the two studies due to the different methods employed and the different datasets considered. Specifically:

1. In Levin and Rödenbeck (2007), each region of interest (upper Rhine valley and the Black Forest) has $^{14}CO_2$ measurements at exactly one site (Heidelberg and Schauinsland) constraining its fossil fuel emissions. In comparison, all the regions for which we have presented optimistic conclusions (such as the United States, Eastern US, or even smaller regions like the NY-NJ-PA tri-state area) have more than one site in and upwind of the region for our "NRC 5000" coverage scenario. Even for the 2010 coverage, large regions such as the Eastern US or the United States (for which monthly fluxes are estimated to within 5% of the "truth") are covered by multiple sites measuring $^{14}CO_2$. The denser coverage in our study is consistent with a more optimistic conclusion compared to Levin and Rödenbeck (2007).

2. Levin and Rödenbeck (2007) had a best-case sampling frequency of once every two weeks at each site. On the other hand, for the NRC 5000 scenario, we have two samples per week at tower sites and one per week at flask sites. This is significantly more frequent than what Levin and Rödenbeck (2007) had, which is consistent with our more optimistic conclusions.

**References**

Bousserez, N., Henze, D. K., Perkins, A., Bowman, K. W., Lee, M., Liu, J., Deng, F., and Jones, D. B. A.: Improved analysis-error covariance matrix for high-dimensional variational inversions: application to source estimation using a 3D atmospheric transport model, Quarterly Journal of the Royal Meteorological Society, pp. n/a—-n/a, doi:10.1002/qj.2495, URL `http://dx.doi.org/10.1002/qj.2495`, 2015.

Chevallier, F., Maksyutov, S., Bousquet, P., Bréon, F.-M., Saito, R., Yoshida, Y., and Yokota, T.: On the accuracy of the CO2 surface fluxes to be estimated from the GOSAT observations, Geophysical Research Letters, 36, n/a—-n/a, doi:10.1029/2009GL040108, URL `http://dx.doi.org/10.1029/2009GL040108`, 2009.

Hungershoefer, K., Breon, F.-M., Peylin, P., Chevallier, F., Rayner, P., Klonecki, A., Houweling, S., and Marshall, J.: Evaluation of various observing systems for the global monitoring of $CO_2$ surface fluxes, Atmospheric Chemistry and Physics, 10, 10 503–10 520, doi:10.5194/acp-10-10503-2010, URL `http://www.atmos-chem-phys.net/10/10503/2010/`, 2010.

Levin, I. and Rödenbeck, C.: Can the envisaged reductions of fossil fuel CO2 emissions be detected by atmospheric observations?, Naturwissenschaften, 95, 203–208, doi:10.1007/s00114-007-0313-4, URL `http://dx.doi.org/10.1007/s00114-007-0313-4`, 2007.

Liu, J., Bowman, K., Lee, M., Henze, D., Bousserez, N., Brix, H., Collatz, G. J., Menemenlis, D., Ott, L., Pawson, S., Jones, D., and Nassar, R.: Carbon monitoring system flux estimation and attribution: impact of ACOS-GOSAT $XCO_2$ sampling on the inference of terrestrial biospheric sources and sinks, Tellus B, 66, URL `http://www.tellusb.net/index.php/tellusb/article/view/22486`, 2014.

Meirink, J. F., Bergamaschi, P., and Krol, M. C.: Four-dimensional variational data assimilation for inverse modelling of atmospheric methane emissions: method and comparison with synthesis inversion, Atmospheric Chemistry and Physics, 8, 6341–6353, doi:10.5194/acpd-8-12023-2008, URL `http://www.atmos-chem-phys-discuss.net/8/12023/2008/`, 2008.

Nassar, R., Sioris, C. E., Jones, D. B. A., and McConnell, J. C.: Satellite observations of CO2 from a highly elliptical orbit for studies of the Arctic and boreal carbon cycle, Journal of Geophysical Research: Atmospheres, 119, 2654–2673, doi:10.1002/2013JD020337, URL `http://dx.doi.org/10.1002/2013JD020337`, 2014.

---

## Author Comment (AC2) · 6 Mar 2016

**Response to Dr. Felix Vogel**

Sourish Basu, John B. Miller, Scott Lehman

March 6, 2016

We thank Dr. Vogel for a prompt and favorable review, and for the helpful questions he raised. Please find below our responses. Dr. Vogel's comments are in *blue italics*, our responses are in normal text. Portions of a response that refer to the main text are in "double quoted red".

*Page 3 equation (1b) Please consider that the mass-balance for 14CO2 is only valid for d14C not D14C. The author discuss the issue of d13C corrections impacting D14C this confusion ca be avoided putting the mass balance for 14CO2 and then mentioning the assumptions made to arrive at a mass balance for D14C. e.g. `https://journals.uair.arizona.edu/index.php/radiocarbon/article/downloadSuppFile/16347/212`. The impact of the approximation in (1b) seems negligible.*

Dr. Vogel is correct that strictly speaking the mass balance of $^{14}CO_2$ only yields an equation in terms of $\delta^{14}CO_2$, and certain assumptions must be made about the relative fractionation of $^{13}CO_2$ and $^{14}CO_2$ to arrive at our equation (1b) in terms of $\Delta^{14}CO_2$. We referred to these assumptions in the sentence immediately after equations (1), viz. "$\Delta_{atm}$ is the isotope signature of $^{14}CO_2$ in the atmosphere expressed in $\Delta$ notation, which includes corrections for mass dependent isotopic fractionation between reservoirs and radioactive decay between the times of sample collection and measurement, such that the quantity $\Delta^{14}CO_2$ is conserved in time." Since the (approximate) mass balance equation in terms of $\Delta^{14}CO_2$ has been covered in previous literature (e.g., Miller et al. (2012)), we did not re-derive it in the manuscript. Indeed, as Dr. Vogel says, the impact of those approximations on equation (1b) is small.

*Page 16/17 conclusions The authors briefly discuss the potential impacts of model transport errors (investigated in section 4.3) and the added value of measurements of auxiliary species. Would you be able to advise on how much more model improvement is needed i.e. should this be an equally/less/more important part of developing the suggested future emission monitoring system?*

While the need for model improvements is clear, Dr. Vogel's question is difficult to answer quantitatively within the present study. One of the two transport models we used for the "imperfect transport" OSSE, namely TM5 EI, was demonstrably biased (figures 2 and 3 in the manuscript), and therefore the difference between "NRC 5000 (EI)" and "Truth" in figure 11 is not an accurate measure of the error from a state-of-the-art unbiased transport model. Even our "better" transport model, TM5 EIC, had a noticeable bias in the vertical profile of $SF_6$ in figure 3. Therefore, at a minimum, transport models need to be improved until they agree with observed atmospheric gradients of passive tracers like $SF_6$ with reasonably well known sources. So to answer Dr. Vogel's question qualitatively, improving atmospheric transport models, especially over continental regions (which

are the primary sources of fossil fuel $CO_2$), needs to be as important as setting up an observational network to monitor future emissions.

**References**

Miller, J. B., Lehman, S. J., Montzka, S. a., Sweeney, C., Miller, B. R., Karion, A., Wolak, C., Dlugokencky, E. J., Southon, J., Turnbull, J. C., and Tans, P. P.: Linking emissions of fossil fuel $CO_2$ and other anthropogenic trace gases using atmospheric $^{14}CO_2$, Journal of Geophysical Research, 117, D08 302, doi:10.1029/2011JD017048, URL http://doi.wiley.com/10.1029/2011JD017048, 2012.